# Changes in Expression of the Genes for the Leptin Signaling in Hypothalamic-Pituitary Selected Areas and Endocrine Responses to Long-Term Manipulation in Body Weight and Resistin in Ewes

**DOI:** 10.3390/ijms21124238

**Published:** 2020-06-14

**Authors:** Dorota Anna Zieba, Weronika Biernat, Malgorzata Szczesna, Katarzyna Kirsz, Justyna Barć, Tomasz Misztal

**Affiliations:** 1Department of Animal Nutrition and Biotechnology, and Fisheries, Faculty of Animal Sciences, University of Agriculture in Krakow, 31-120 Krakow, Poland; weronikaa.biernat@gmail.com (W.B.); malgorzata.szczesna@urk.edu.pl (M.S.); katarzyna.kirsz@urk.edu.pl (K.K.); justyna.barc@urk.edu.pl (J.B.); 2Department of Animal Physiology, The Kielanowski Institute of Animal Physiology and Nutrition, Polish Academy of Sciences, 01-224 Jablonna, Poland; t.misztal@ifzz.pl

**Keywords:** leptin, LRb, SOCS-3, RSTN, sheep, nutrition

## Abstract

Both long-term undernutrition and overnutrition disturb metabolic balance, which is mediated partially by the action of two adipokines, leptin and resistin (RSTN). In this study, we manipulated the diet of ewes to produce either a thin (lean) or fat (fat) body condition and investigated how RSTN affects endocrine and metabolic status under different leptin concentrations. Twenty ewes were distributed into four groups (*n* = 5): the lean and fat groups were administered with saline (Lean and Fat), while the Lean-R (Lean-Resistin treated) and Fat-R (Fat-Resistin treated) groups received recombinant bovine resistin. Plasma was assayed for LH, FSH, PRL, RSTN, leptin, GH, glucose, insulin, total cholesterol, nonesterified fatty acid (NEFA), high-density lipoprotein (HDL)-cholesterol, low-density lipoprotein (LDL)-cholesterol and triglycerides. Expression levels of a suppressor of cytokine signaling (SOCS-3) and the long form of the leptin receptor (LRb) were determined in selected brain regions, such as the anterior pituitary, hypothalamic arcuate nucleus, preoptic area and ventro- and dorsomedial nuclei. The results indicate long-term alterations in body weight affect RSTN-mediated effects on metabolic and reproductive hormones concentrations and the expression of leptin signaling components: LRb and SOCS-3. This may be an adaptive mechanism to long-term changes in adiposity during the state of long-day leptin resistance.

## 1. Introduction

Leptin, an adipocyte hormone that is secreted in proportion to body fat levels, inhibits food intake to provide feedback control of fat mass, facilitating maintenance of a constant body weight (BW) [1]. Leptin receptors are expressed in several brain regions, with the main site of action being the hypothalamic nuclei, mainly the arcuate nucleus (ARC) [2,3]. Paradoxically, in common forms of obesity, paradoxically, as fat mass increase, circulating leptin concentrations also increase but fail to suppress food intake. The origin of leptin resistance is still not fully understood, and solving this puzzle is widely considered the key to both understanding how obesity develops and identifying effective therapeutic interventions.

In our most recent studies, we tested a hypothesis that leptin resistance was a consequence of the hyperleptinemia associated with increasing fat depots and action of resistin (RSTN) [4,5]. The first significant evidence for that hypothesis was provided by Asterholm’s group, who showed that chronically elevated RSTN concentrations lead to leptin resistance and reduced the expression of leptin signaling components, namely, suppressor of cytokine signaling-3 (SOCS-3) in the hypothalamic nuclei [6]. This supports our earlier suggestions that an increase in the expression of SOCS-3 in the ARC may be an executive player in the development of leptin resistance in sheep [7,8,9]. Another confirmation supporting our hypothesis was provided by a study from Friedman’s group demonstrating that leptin-deficient animals in which BWt was normalized by chronic low-level leptin infusion remained responsive to leptin, even when obesity was induced by a high-fat diet (HFD) [10]. These data showed that HFD and obesity are insufficient to cause leptin resistance and that hyperleptinemia is required to induce this state. Recently, we found proof supporting the role of RSTN, an adipokine that is expressed and produced in ruminant white adipose tissue, in hyperleptinemia in sheep during in in vitro and in vivo experimental conditions [4]. Moreover, RSTN was determined to be engaged in modulation of central, photoperiod-driven changes in the expression of a long form of leptin receptor b (LRb) and SOCS-3 in the anterior pituitary (AP) and selected hypothalamic nuclei [4]. Earlier, in 2018, our team presented that in seasonally breeding ovariectomized ewes with estradiol replacement action of RSTN on the level of AP appeared to be firmly dose- and photoperiod-dependent. RSTN modulated reproductive hormone secretions: luteinizing hormone (LH), follicle stimulating hormone (FSH) and prolactin (PRL) [11]. Thus, it has been suggested that RSTN is involved in the regulation of seasonal reproduction in sheep [11].

Nutrition can alter the expression of many peptides and neuropeptides that regulate food intake. However, most experiments in animals have investigated acute changes with fasting [12,13], and the effects of long-term alterations in body weight and adiposity still require complete explanation. In that subject, some improvement has been made using ruminant models, and the expression of leptin in the white adipose tissue and neuropeptide Y in brain tissue was determined after long-term food deprivation and revealed differences between fasted, normal-fed and fat animals [14,15]. On the other hand, impairment of leptin secretion and/or action results in weight gain by sending an inappropriate signal to the brain, which consequently reduces satiety response. Leptin secretion and circulating leptin concentrations can be manipulated by the choice of diet (reduces in carbohydrate vs. fat intake), which often causes a lowering leptin concentration [16]. The likely reason behind reduced leptin serum concentrations is dietary fat content (at least 60% of calories from fat), which decreased insulin-mediated glucose metabolism in adipocytes, which is in turn associated with reduced leptin expression and secretion from adipocytes [17,18]. This is because induced lipolysis increases the expression of peroxisome proliferator–activated receptor gamma, which reduces the expression of leptin in adipose tissue [19,20].

Energy balance is tightly coupled with the physiological status of organisms, and matching nutritional reserves to reproductive activity is fundamental to the survival of a species. Therefore, metabolic abnormalities connected with obesity lead to the development of pathophysiological conditions such as polycystic ovary syndrome and metabolic syndrome, and studies indicate that RSTN may play a role in both diseases [21,22]. Leptin and RSTN both regulate metabolism and activate the same central signaling pathway in the hypothalamus; studies on lean and obese subjects would help to answer some questions concerning molecular aspects of metabolic disturbances with those energy status indicators. Although, increased expression of LRb had previously been reported in chronic undernutrition or a negative energy states in ewes [23], it was not determined whether changes in the expression of transcripts of LRb and SOCS-3 occur in animals that are maintained in an obese state. In the current study, we used two different metabolic/nutritional sheep models (lean and fat) to test the hypothesis that alternations in body weight affect RSTN-mediated effects on leptin signaling component expressions (LRb and SOCS-3) at the level of the selected hypothalamic nuclei (ARC, medio-basal hypothalamus comprising of ventro-medial/dorso-medial (VMH/DMH) nuclei, preoptic area (POA) and AP, brain areas engaged directly in the regulation of metabolism and leptin action. Furthermore, the endocrine (reproductive and metabolic) indicators in lean and fat sheep with alternate leptin concentrations (low and high) were investigated.

## 2. Results

### 2.1. BW and Hormonal Status

There were significant (*p* < 0.01) differences in BW between the lean and fat groups, with the lean ewes weighing 41.2 0 ± 0.92 and the fat ewes weighing 78.1 ± 1.78 kg (Figure 1). Abdominal fat weight was significantly (*p* < 0.01) less in lean animals 0.2 ± 0.02 kg determined postmortem, than in fat animals 5.3 ± 0.4 kg.

The mean circulating concentration of estradiol was 3.6 ± 0.3 pg/mL in blood samples collected during the pretreatment period in all groups of sheep.

The concentrations of circulating RSTN in the lean group of sheep were lower (*p* < 0.05) than those observed in fat animals (Table 1). The concentrations of leptin, insulin and nonesterified fatty acids (NEFAs) were lower (*p* < 0.01) in the lean groups than in the fat groups of ewes (Table 1). No significant changes were noticed in the concentrations of glucose, total cholesterol, high-density lipoprotein (HDL) and low-density lipoprotein (LDL; *p* = 0.6–0.8; Table 1). The plasma concentrations of triglycerides were higher (*p* < 0.01) in the Lean-R (Lean-Resistin treated) sheep than in the lean group of sheep (Table 1).

The plasma concentrations of LH, FSH, PRL and growth hormone (GH) were altered in accordance with changes in body weights and adiposity in the lean and fat groups of animals (Table 2). In particular, LH, FSH and PRL concentrations were lower (*p* < 0.01 for LH; *p* < 0.05 for FSH and PRL) in lean sheep (Table 2) compared to fat sheep. Food restriction increased (*p* < 0.001) concentrations of circulating GH. The secretory profiles of LH and GH were characterized using previously reported pulse analysis techniques [11]. Pulse amplitudes of LH were higher (*p* < 0.05) in fat ewes than in lean ewes, with no changes in pulse frequency; however, GH pulse amplitude and frequency did not significantly differ between the lean and fat groups of ewes.

The injection of 5.0 µg/kg BW rbresistin increased (*p* < 0.05) endogenous RSTN concentrations in Lean-R (Lean-Resistin treated) and Fat-R (Fat-Resistin treated) sheep compared to nontreated ewes (Table 1). Concentrations of leptin and insulin after rbresistin treatment were higher (*p* < 0.001) in Fat-R sheep than in the fat and Lean-R groups of ewes (Table 1). Glucose concentrations were lower (*p* < 0.05) in Lean-R sheep after exogenous rbresistin injection compared to lean and Fat-R groups and higher (*p* < 0.05) in Fat-R vs. fat ewes (Table 1). Exogenous rbresistin lowered (*p* < 0.01) plasma NEFA concentrations in the Fat-R vs. fat group and increased (*p* < 0.01) vs. Lean-R group (Table 1). Triglyceride and HDL fraction concentrations were lower (*p* < 0.05) after RSTN treatment in the Lean-R vs. lean and Fat-R groups (Table 1).

Recombinant bovine resistin increased (*p* < 0.01) LH concentrations and pulse amplitudes in Lean-R and Fat-R compared to levels in nontreated groups (Table 2). The FSH concentrations were higher (*p* < 0.05) after exogenous RSTN treatment in the Lean-R and Fat-R groups than in the lean and fat groups (Table 2). Rbresistin injection increased (*p* < 0.01) concentrations of PRL in both the Lean-R and Fat-R groups compared to the nontreated groups of sheep (Table 2). Growth hormone concentrations were lower (*p* < 0.01) in the Lean-R group vs. lean ewes and in the Fat-R sheep compared to concentrations in the fat ewes (Table 2).

### 2.2. Expression of mRNA LRb and SOCS-3 in Selected Brain Areas

There were no significant changes in the pituitary leptin receptor or transcript level between the lean and fat groups (*p* = 0.07). Rbresistin decreased LRb expression in AP 3.0-fold (*p* < 0.001) in Fat-R compared to the expression noted in Lean-R and 2.0-fold (*p* < 0.001) vs. LRb expression observed in the fat and lean groups (Figure 2).

Leptin Rb transcripts were detected at varying levels in all examined hypothalamic tissues: the ARC, POA and VMH/DMH in the nontreated (lean and fat) and RSTN-treated (Lean-R and Fat-R) groups of sheep, with the exception of the undetectable level of expression in the POA in the Lean-R group and the VMH/DMH in the Fat-R group of ewes (Figure 2).

Within the ARC, LRb mRNA levels were higher (*p* < 0.001) in the fat vs. lean groups of sheep; after rbresistin injection, an increase in the level of LRb transcripts was noted in the Lean-R and Fat-R groups (*p* < 0.001) vs. lean ewes.

Transcripts of LRb in the POA area increased 8-fold in the Fat-R (*p* < 0.001) vs. the level expression noted in the fat and lean group.

The expression of LRb in the VMH/DMH showed no significant differences among the three experimental groups (Figure 2).

There was a 35-fold higher (*p* < 0.001) level of SOCS-3 expression in the fat group than in the lean group in the pituitary (Figure 3). Furthermore, SOCS-3 transcript levels in the Fat-R and Lean-R groups increased 35-fold and 45-fold (*p* < 0.001), respectively, compared to those levels noted in the lean sheep.

The expression of SOCS-3 was detectable in all examined hypothalamic tissues during the experiments (Figure 3). A high difference (*p* < 0.001) in the level of SOCS-3 expression was observed between the lean and fat groups in the ARC area (Figure 3). Levels of SOCS-3 mRNA increased significantly (*p* < 0.001) after the injection of rbresistin in the Fat-R (45-fold) and Lean-R (40-fold) groups of ewes compared to the levels noted in the lean group (Figure 3).

In the POA, after rbresistin treatment, a high increase (*p* < 0.001) in SOCS-3 expression was noted in Lean-R sheep vs. the level detected in lean sheep (Figure 3). The same effect of rbresistin treatment (*p* < 0.001) was noted in Fat-R vs. fat sheep. The transcript level of SOCS-3 was significantly higher (*p* < 0.001) in Fat-R than in lean animals (Figure 3).

No differences (*p* ≥ 0.05) were found in SOCS-3 transcript levels in the VMH/DMH between lean and fat sheep, but rbresistin treatment increased (*p* < 0.001) the transcript level of cytokine suppressors by 12-fold in the Fat-R and 25-fold in the Lean-R groups vs. lean ewes (Figure 3).

## 3. Discussion

The results of the present study and previous studies [4] indicated not only the existence of leptin resistance on the level of the hypothalamus but also RSTN-stimulated hyperleptinemia in sheep (both Lean-R and Fat-R sheep groups). Furthermore, long-term alterations in BW, especially increased adiposity, affect the effects of RSTN on the transcript expression of LRb/SOCS-3 in selected hypothalamic nuclei and AP.

The central action of appetite-regulated peptides is mediated primarily by the balance between anorexigenic neurons expressing leptin, RSTN, pro-opiomelanocortin and orexigenic neurons expressing NPY and ghrelin. Numerous studies suggest that adipocyte-derived RSTN and leptin directly regulate both feeding and peripheral metabolism through, so far for RSTN undefined hypothalamic-mediated mechanisms [24,25]. Hypothalamic neurons participate in the detection of circulating and locally synthesized peptides carrying information relevant to the body energy status, while the ARC plays a crucial role in the integration of this information. They lie directly above median eminence one of the “periventricular” areas where the blood–brain barrier is modified. This allows even larger peripheral peptides, such as leptin or/and RSTN, to pass through the peripheral signal to the ARC and hence to different nuclei (VMH/DMH) and other brain regions, such as the POA or AP [4,25]. Arcuate neurons are characterized by a large number of receptors for peripheral peptides, such as leptin (LRb), insulin (IR) and RSTN (TLR4) [26]. Domestic animals, unlike rodents, are not nocturnal, so the important diurnal patterns of behavior and physiology and BW of sheep are more similar to those of humans, and they allow the definition of precise endocrine parameters through serial blood sampling. Data collected from the experiments indicated that in Lean-R sheep, RSTN treatment induced a marked increase in triglyceride levels, indicating an overall activation of triglyceride uptake and fatty acid synthesis in the liver under these reduced food supply long-term conditions. This is in agreement with the results of a study of RSTN-treated rats [25]. Consistent with these changes, there was a marked increase in triglycerides in Fat-R sheep, a phenomenon also noted in rat studies [27]. On the other hand, nutrition alternation impacted NEFA, increasing its concentration in the fat sheep, as well as insulin concentrations, HDL fraction and whole cholesterol. In the fat group of sheep, high insulin concentrations (hyperinsulinemia) increased adipose tissue lipolysis and led to impaired NEFA uptake and an increase in plasma NEFA and triglyceride concentrations. Analysis of NEFA provides information about the individual’s lipid metabolism and pathogenesis of obesity (fat sheep model). RSTN infusion results in major changes in the peripheral expression of lipogenic factors in a tissue-specific and nutrition-dependent manner. In rats, fed-state RSTN infusion was associated with induced expression of fatty acid synthesis enzymes and proinflammatory cytokines in the liver, whereas its administration in the fasted state did so in white adipose tissue [27]. In Fat-R sheep, RSTN treatment decreased the NEFA concentration and HDL fraction vs. fat group, and those effects were probably related to the RSTN-induced increase in acetyl-CoA carboxylase expression, as demonstrated in rat experiments [27]. Of interest, RSTN also exerts a profound nutrition-dependent inhibitory effect on hypothalamic fatty acid metabolism, as indicated by increased phosphorylation levels of both AMP-activated protein kinase and its downstream target acetyl-coenzyme A carboxylase, associated with decreased expression of fatty acid synthase in the VMH [28]. During the experiments, plasma concentrations of glucose were slightly different between nutritional groups. Sheep do not absorb glucose from their diet and utilize volatile fatty acids as the primary dietary energy source; thus, glucose concentrations were maintained via gluconeogenesis. A study by Park et al. [29] in rats reported the important concept that central leptin and RSTN regulate peripheral glucose homeostasis via independent pathways in which RSTN downregulates leptin signaling. Potential physiological consequences of such a mechanism are that in pathological situations, the annulling effect of RSTN on leptin signaling favors the development of hyperinsulinemia to overcome insulin resistance during obesity. Hyperinsulinemia can then promote the redirection of glucose and lipid flux to adipocytes, preventing the well-known detrimental effect on muscles and the liver.

Overall, our results indicate that RSTN controls peripheral lipid metabolism in sheep and is probably mediated by the activation of de novo lipogenesis in the liver and lipolysis in white adipose tissue, as demonstrated by the results of an in vitro study in bovine adipose tissue explants by Reverchon et al. [30]. Alternations in body weight, which are maintained for an extended period of time, allow some energy balance peptides to adapt to nutritional status. One of them in sheep is leptin, whose concentration reflects not only body fat reserves but also adaptation to external environmental conditions, such as the creation of a state of reversible seasonal central resistance [4,7,31]. This adaptation was observed by us in the present study conducted during LD hypothalamic leptin insensitivity and extremely high leptin concentrations after RSTN treatment, indicating hyperleptinemia.

Chronic undernutrition can markedly increase the secretion of GH in both ruminant species and humans [32], whereas in rodents, serum concentrations of GH decrease during undernourished states [33]. The present study showed that RSTN consecutively decreased the plasma concentration of GH in both nutritionally altered animals. A study of rats suggested that RSTN might regulate GHRH or SRIF secretion to obtain GH release, or as has been reported previously, RSTN could act directly over somatotropes to regulate GH release [34].

Data from many studies in ruminants proved that leptin regulates reproductive processes depending on metabolic and photoperiodic conditions, and our experiments demonstrated that RSTN is engaged in mechanisms of those regulations as well [11,31]. Seasonal breeding is a remarkable adaptive feature that allows sheep to optimize their survival and reproductive success. Our group indicated for the first time that RSTN is involved in the regulation of pituitary hormone secretions and that those effects were differentially mediated during the LD and short-day seasons [11]. RSTN regulated the release of reproductive hormones from the AP in a dose- and day length-dependent manner [11,35]. The present experiments demonstrated that rbresistin significantly increased the mean concentrations of LH and FSH and the mean circulating concentration of PRL. These results are in agreement with our previous studies, which we conducted during lengthening days using low and high doses of exogenous RSTN [11]. Leptin signaling does not work properly during LD, and RSTN strengthens the failure of leptin signaling via rising expression of SOCS-3, which is why lower concentrations of gonadotrophins were noted presently. Furthermore, experiments determined the lower amplitude of LH pulses, which confirmed that RSTN is able to act centrally at the level of the hypothalamus and modulate GnRH secretion. Interestingly, a study of rats confirmed that RSTN is present in the mediobasal hypothalamus and acts centrally on metabolism and inhibits leptin-mediated STAT3 phosphorylation in hypothalamic slices in organotypic culture [6], so it cannot be excluded that RSTN can directly affect the hypothalamic–pituitary–gonadal axis. On the one hand, RSTN receptors have been suggested to be present in AP cells in rats [36]. On the other hand, the mechanism by which leptin, RSTN and kisspeptin interact is likely involved. In mammals, including sheep, two major populations of kisspeptin-synthesizing neurons exist in the POA and ARC of the hypothalamus. Kisspeptin is a potent secretagogue for LH release, which appears to be GnRH-dependent [37]. A direct effect of kisspeptin on the pituitary gonadotropes to stimulate LH release appears unlikely because kisspeptin administration failed to stimulate gonadotropin secretion in hypothalamo-pituitary disconnected ewes [38]. Among the intermediate neurons that possess leptin receptors and transmit adipose tissue/leptin messages to GnRH-producing neurons are kisspeptin neurons. The importance of kisspeptin is that it forms a critical link between energy homeostasis and reproduction [39]. Kisspeptin concentrations are suppressed by food restriction, such as a 72 h fast, which may explain the disruption of the reproductive axis during a negative energy balance. In Lean-R sheep leptin, LH and FSH concentrations and SOCS-3 expression in POA and AP were higher compared to lean ewes. It seems to be possible that the RSTN affected reproductive parameters and leptin action in the POA by acting through kisspeptin neurons, but this effect must be confirmed in future studies.

As the present experiments proved, the highest expression of LRb was localized to the ARC nuclei and was dependent not only on the nutritional status of the females, with predominant expression in lean sheep, but also on RSTN treatment. As noted, the high expression of the SOCS-3 transcript was observed in exactly the same area in the ARC as LRb, which explained the existence of leptin central resistance resulting from the action of SOCS-3 as a factor interfering with the transmission of information from leptin. These results have conclusively confirmed the observations made by our group during previous experiments showing day length- (photoperiodically) and RSTN-dependent leptin insensitivity [4]. In addition, they emphasized the existence of another element involved in this phenomenon, which is the action of RSTN depending on the metabolic status of the organism. Thus, the main work hypothesis regarding the impact of long-term metabolic manipulation in sheep on RSTN-mediated effects on leptin signaling components (LRb and SOCS-3) at the level of the selected hypothalamic nuclei was confirmed.

The expression of LRb transcripts in the VMH/DMH was not significantly different between sheep groups; however, a trend close to significance (*p* = 0.06) was observed in the Lean-R group. This is not surprising since one of the primary factors associated with an increase in the number of LRb in the VMH of ewes is malnutrition [23]. RSTN appeared to be another factor that acted in this hypothalamic area and affected the expression of the long form of leptin receptor. The present study indicated that RSTN is able to increase leptin resistance by increasing leptin concentration and integrating leptin signaling components.

Thus, in addition to the pivotal role that RSTN plays in weight homeostasis by coordinating energy intake and expenditure, these findings reveal a new putative role of RSTN in regulating dynamic afferent hormonal signaling that provides hypothalamic feedback. Since leptin and RSTN are implicated in the etiology of obesity and metabolic syndrome, it is important to characterize the hormonal cross talk between the peripheral and central loci at the cellular and molecular levels. Similarly, the precise route of efferent outflow activated by central leptin and RSTN targets to modulate reproductive hormone secretion in the hypothalamus warrants delineation.

In conclusion, we have shown that alterations in BW influence the effects of RSTN on metabolism, reproductive parameters and the leptin signaling pathway in sheep. Our results suggest that nutrition-induced alterations in the expression of LRb and SOCS-3 can be linked to the endocrine and metabolic adaptations that occur in response to long-term changes in adiposity. RSTN appears to be another adipokine, in addition to leptin, that is involved in the regulation of physiological processes in sheep. RSTN regulates the release of reproductive hormones from the pituitary and metabolic processes of sheep.

## 4. Materials and Methods

The Second Local Ethics Committee on Animal Testing in Krakow, Poland, approved all procedures conducted on animals during the experiments (Protocol No. 109/2018).

### 4.1. Animals

The studies were conducted at the Experiment Station of the Department of Animals Nutrition and Biotechnology, and Fisheries at the University of Agriculture in Krakow (longitude: 19°57 E, latitude: 50°04 N) during the LD season (between April and May). A total of twenty female Polish Longwool sheep, a breed that exhibits strong reproductive seasonality, were prepared for experiments (ovariectomized with estrogen replacement) as described in detail by Biernat et al. [11] to prevent cyclic alterations in the secretion of gonadal steroids that could create differences between animals, especially those with different body conditions. Animals 2–3 years of age were weighed at the start of the protocol (mean ± SD) 60 ± 3.1 kg and housed under natural photoperiodic and thermoperiodic conditions. The adiposity score, also known as the body condition score (BCS), was estimated by palpation of the prominence and degree of cover of the spinous and transverse processes of the anterior lumbar vertebrae. The average BCS was 3.2 ± 0.2 on a scale of 1–5 (1 = emaciated; 5 = obese) [40].

### 4.2. Experimental Design

Animals were randomly assigned into two groups that were provided altered diets over 5 months: a food-restricted diet (lean, *n* = 10) and a high-energy diet designed to increase BW (fat, *n* = 10). When food restriction was applied for extended periods, the sheep were housed in groups, and lean animals received approximately 400 g of pasture hay/day supplemented with straw ad libitum as a filler. Those sheep were managed very carefully, as competition for food could occur, and some ewes may be more severely compromised than others. The objective of the adiposity score was to alter the BCS of animals to 2. The fat animals received pasture hay ad libitum plus a dietary supplement of approximately 1 kg of lupin grain/animal/week, which increased adipose deposition. The sheep were weighed every second week, and the target weights were reached by 4 months, after which the diets were maintained for another month. The nutrient requirements of sheep and the exact compositions of the diets were determined based on Institut National de la Recherche Agronomique (INRA), [41].

### 4.3. Animal Treatment

**Experiment** **1.**
*Effects of alterations in BW and rbresistin treatment on plasma hormone concentrations in lean and fat sheep.*


During LD, in April/May, ewes were placed frequently into carts according to a previous report to avoid stress during the experiment [7]. Carts were constructed of wood with solid floors, and they allowed animals to stand or lie down freely during sampling procedures and have visual contact with other sheep. In the morning on the day of each experiment, sheep were fitted with jugular vein catheters (central and peripheral venous catheters, Careflow^TM^, Argon, Billmed Sp. z o.o., Warsaw, Poland) for intensive blood sampling to determine the endocrine profile and metabolic status of the sheep before treatment with rbresistin (pretreatment status). Blood samples (5 mL) were collected at 10-min intervals for 4 h. At the end of intensive blood sampling, a washout period of three days elapsed before ewes were allocated into treatment groups.

Sheep were assigned to one of four treatment groups (*n* = 5/group/treatment). The experimental groups were as follows: Lean (*n* = 5) and Fat (*n* = 5) groups injected with saline (5.0 mL) and Lean-R (Lean-Resistin treated; *n* = 5) and Fat-R (Fat-Resistin treated; *n* = 5) groups injected intravenously one time with rbresistin, 5.0 µg/kg BW (5.0 mL). The rbresistin dose was chosen based on our previous studies [4,11]. Recombinant bovine resistin was purchased from CliniSciences (Nanterre, France). In the morning of the beginning of the experiment, sheep from each group were fitted with jugular vein catheters for intensive blood sampling as described previously. At the beginning of the experiment (time 0), blood was collected through the catheter, and rbresistin was then injected through the same catheter. Blood samples (5 mL) were collected at 10-min intervals over 4 h.

After both blood sample collections, samples were dispensed into tubes containing 150 µL of a solution containing heparin (10 000 IU/mL), *Heparinum natricum* purchased from Polfa S.A. (Warsaw, Poland) and 5% (*w/v*) EDTA (ethylene diamine tetraacetic) purchased from Polfa S.A. (Warsaw, Poland) and placed on ice immediately. Plasma was separated by centrifugation and stored at −20 °C until analyses. Analyses included plasma concentrations of LH, FSH and PRL to determine reproductive conditions; RSTN, leptin, growth hormone (GH), glucose, insulin, total cholesterol, nonesterified fatty acids (NEFA), HDL-cholesterol, LDL-cholesterol fractions and triglycerides to provide indicators of the metabolic status of sheep. Plasma LH and GH concentrations were measured in every sample, and 100 µL of plasma from each sample was pooled for the measurement of RSTN, leptin, glucose, insulin, total cholesterol, NEFA, HDL-cholesterol, LDL-cholesterol and triglycerides.

**Experiment** **2.**
*Alterations in body weight and rbresistin treatment on the expression of LRb and SOCS-3 transcripts in selected brain areas in lean and fat sheep.*


At the end of the second intensive blood sampling, a washout period of one week elapsed before ewes were allocated into the same treatment groups. One hour after saline/rbresistin infusion (a dose of 5.0 µg/kg BW), the animals were humanely euthanized by captive bolt stunning. Brains with the infundibulum remaining untouched were quickly removed from the skulls of all sheep and frozen on dry ice. Samples of the AP, the hypothalamic specific areas: ARC, the VMH/DMH and the POA were aseptically dissected 10–15 min postmortem. The selected brain regions were collected by removing a tissue block encompassing the hypothalamic–infundibular complex, followed by transection into two halves. An anterior coronal cut was made 3–5 mm rostral to the optic chiasm, and a posterior coronal cut was made, which contained approximately one-third of the mamillary body. A longitudinal cut parallel to the ventral surface of the brain 2–3 cm dorsal to the anterior commissure followed. At the same time, the pituitary was harvested from the *sella turcica*. The probability of contamination caused by transferring tissue between samples was eliminated using separate sterile tools to dissect the chosen area. Samples of brain tissue were rinsed in phosphate-buffered saline (PBS; Laboratory of Vaccines, Lublin, Poland), snap-frozen in liquid nitrogen, and then stored at −80 °C until real-time PCR analysis.

Following euthanasia, measurements were made for abdominal fat weight.

### 4.4. Hormones and Metabolites Assays

LH concentrations were measured by a double-antibody (anti-ovine LH and anti-rabbit γ-globulin antisera) radioimmunoassay (RIA) developed by Stupnicki and Madej [42] with a bovine LH standard (NIH-LH-B6). The assay sensitivity was 0.3 ng/mL, and the intra- and interassay coefficients of variation (CVs) were 6.7% and 11.2%, respectively. Pulse parameters of LH secretion were calculated using the Pulsar Computer Program [43]. Analyses were performed for each sheep, and they included the entire sampling period. Plasma FSH concentrations were measured by RIA using anti-ovine FSH and FSH standards, which were generously supplied by Dr. L. E. Reichert Jr. (Tucker Endocrine Research Institute LLC, Stone Mountain, USA). The assay sensitivity was 1.6 ng/mL, and intra- and interassay CVs were 3.4% and 9.2%, respectively. Plasma concentrations of PRL were assayed by the double antibody method using anti-ovine PRL and anti-rabbit-γ-globulin antisera [44]. The assay sensitivity was 2 ng/mL, and the intra- and interassay CVs were 7.0% and 13.0%, respectively. Plasma concentrations of GH were assayed by a radioimmunological, double antibody method using anti-bovine GH and anti-rabbit-*γ*-globulin antisera, as previously described by Dvorak et al. [45]. Bovine GH (NIDDK-GH-B-1003A) served as a reference standard. The assay sensitivity was 0.4 ng/mL, and the intra- and interassay coefficients of variation were 5.3% and 9.31%, respectively. Estradiol concentrations were determined using commercially available enzyme immunoassay (EIA) kits (DRG Instruments GmbH, Marburg, Germany) according to the manufacturer’s instructions. The inter- and intra-assay precision values exhibited CVs of 3.46% and 2.4%, respectively. The assay sensitivity was 1.9 pg/mL. RSTN concentrations were determined using commercially available EIA kits (Cloud-Clone Corp., Katy, TX, USA) according to the manufacturer’s instructions. The inter- and intra-assay precision values exhibited CVs of 6.6% and 2.3%, respectively, and the assay sensitivity was 2.2 pg/mL. Plasma leptin concentrations were determined using commercially available RIA kits (Multispecies Leptin RIA, EMD Millipore Co., Billerica, MA, USA) according to the manufacturer’s instructions. The inter- and intra-assay precision values exhibited CVs of 3.46% and 2.4%, respectively. The assay sensitivity was 0.8 ng/mL. Plasma insulin concentration was assayed using commercially available EIA kits (Mercordia AB, Uppsala, Sweden) according to the manufacturer’s instructions. The inter- and intra-assay precision values exhibited CVs of 2.25% and 3.6%, respectively. The assay sensitivity was 2 mU/L. Glucose was measured using the YSI 2300 STAT PLUS Glucose/Lactate Analyzer (YSI Inc., Yellow Springs, OH, USA). The concentrations of total cholesterol (TC), high-density lipoprotein cholesterol (HDL) and low-density lipoprotein cholesterol (LDL), triglycerides (TG) were determined spectrophotometrically (PowerWaveTM XA, BioTek, Winooski, VT, USA) using colorimetric test kits (Pointe Scientific, Warsaw, Poland) according to the manufacturer’s specifications. The concentration of low-density lipoprotein cholesterol (LDL) was calculated on the basis of Friedewald’s formula (LDL = TC-[HDL + TG/5]). NEFAs were measured using a colorimetric assay kit (Bio Tech Co, Dallas, TX, USA).

### 4.5. Statistics

#### 4.5.1. Statistical Analysis

All data are presented as the mean ± SEM. Data analysis was performed by a series of two-way ANOVAs using SigmaPlot^®^ statistical software (version 11.0; Systat Software Inc., Richmond, CA, USA), preceded by Grubb’s test to identify outliers. All data sets with failed tests of normality and/or equal variance were transformed as natural logarithms. If the main effects or their interactions were significant, the Holm–Sidak test was used as a post-ANOVA test to compare individual means. A *p* value < 0.05 was considered to indicate statistical significance.

#### 4.5.2. Quantitative Real Time RT-PCR Analysis

The mRNA expression of LRb and SOCS-3 was measured using the real-time PCR method. Tissue homogenization was performed with a rotor-stator homogenizer (Omni TH, Omni International, Inc., Kennesaw, GA, USA) and single-use tips (Soft Tissue Omni Tip Plastic Homogenizing Probes, Omni International, Inc.). Total RNA was isolated using a TRIzol reagent (Ambion Inc., Austin, TX, USA) following the manufacturer’s protocol. Incubation of samples at 42 °C for 2 min with gDNA Wipeout Buffer (QuantiTect Reverse Transcription Kit; Qiagen, Hilden, Germany) was used to eliminate contamination of genomic DNA. Subsequently, to obtain samples of cDNAs by reverse transcription, isolates of RNA (1 µg) were incubated with Quantiscript reverse transcriptase and RT primer mix (QuantiTect Reverse Transcription Kit; as above) at 42 °C for 15 min. The reaction was terminated by heating the samples to 94 °C for 3 min. Amplification of each cDNA was performed in triplicate using an Applied Biosystems 7300 Real-Time PCR System, TaqMan Gene Expression Master Mix, 900 nM concentrations of specific primers corresponding to the target/reference genes (sequence detection primers) and 250 nM concentrations of specific probes corresponding to the target/reference genes (TaqMan MGB Probes) supplied by Life Technologies (Foster City, CA, USA). Primers and probes were designed using Primer Express software v. 2.0 (Applied Biosystems; Foster City, CA, USA) and are characterized in Table 3. The thermal profile of the real-time PCR was as follows: (1) 50 °C for 2 min—initial incubation, (2) 95 °C for 10 min—activation of polymerase and (3) 40 cycles with denaturation (95 °C for 15 sec) and annealing/elongation (60 °C for 60 sec). The collected data were recorded with the Applied Biosystem 7300 Real-Time PCR System SDS software.

#### 4.5.3. Quantitative Real Time RT-PCR Data Analysis

The expression levels were calculated using relative quantification (RQ) analysis, and the results were expressed as a function of the threshold cycle (Ct), which is a value corresponding to the fractional PCR cycle number at which the fluorescent signal reached the detection threshold. Data were analyzed using the 2^−ΔΔCt^ method, and Ct values were converted to fold-change RQ values. The RQ values from each gene were used to compare target gene expression across all groups. The mean mRNA expression levels for target genes in each sample were standardized against the expression of a reference gene (cyclophilin; CPH) and expressed relative to the calibrator sample. The variation in the Ct values for CPH among the treatment groups was not significant (*p* > 0.05). The mean ΔCt value for the indicated tissues collected from the lean group was used as a calibrator to compare the changes in target gene expression among all treatment groups in the indicated season.

Differences in the means were compared with SigmaPlot statistical software (version 11.0; Systat Software Inc., Richmond, CA, USA), using all pairwise multiple comparison procedures (Tukey test), preceded by the determination of a significant F-value. Differences were considered statistically significant when *p* < 0.05.

## Figures and Tables

**Figure 1 ijms-21-04238-f001:**
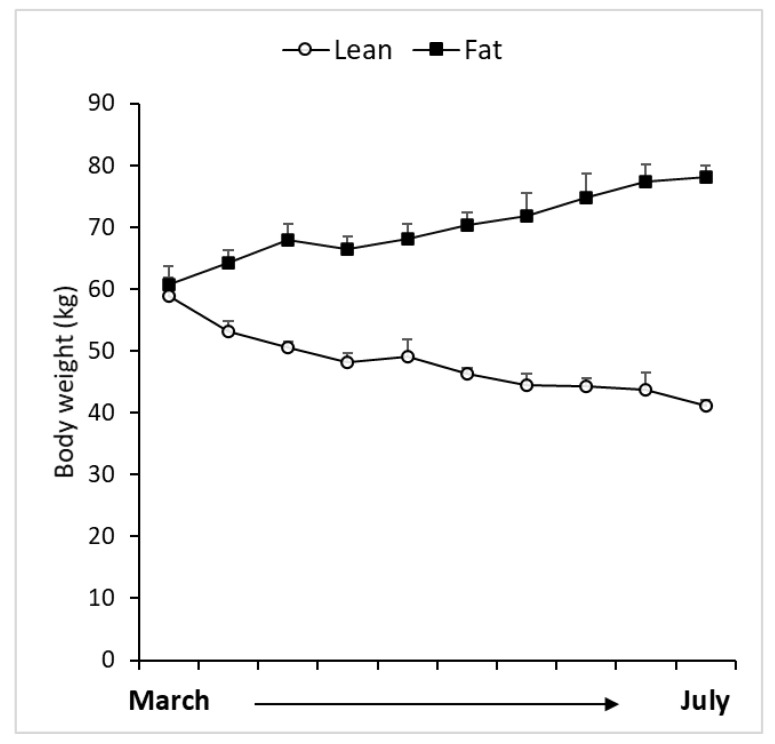
Temporal changes in body weight in lean and fat animals; which were maintained until July (time of experiment).

**Figure 2 ijms-21-04238-f002:**
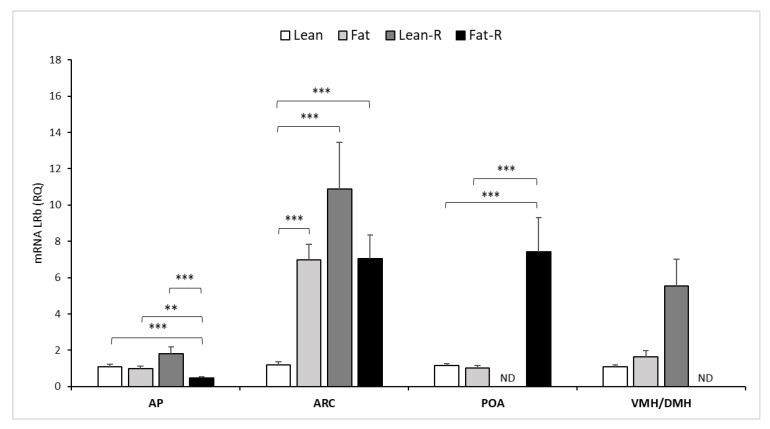
Leptin receptor mRNA expression. The mean mRNA expression (±SEM) of the long form of the leptin receptor (LRb) in the ovine anterior pituitary gland (AP), arcuate nucleus (ARC), preoptic area (POA) and ventro- and dorsomedial nuclei (VMH/DMH) collected during a long-day (LD) photoperiod in nontreated (Lean and Fat) animals and animals treated with recombinant bovine resistin (Lean-R [Lean-Resistin treated] and Fat-R [Fat-Resistin treated]. The expression of LRb mRNA is reported in arbitrary units (RQ) relative to cyclophilin mRNA expression and expressed relative to the calibrator sample. Differences relative to the control or between the other groups are indicated with ** *p* < 0.05, *** *p* < 0.001. Samples in which the expression of the target gene was undetectable are designated with ND.

**Figure 3 ijms-21-04238-f003:**
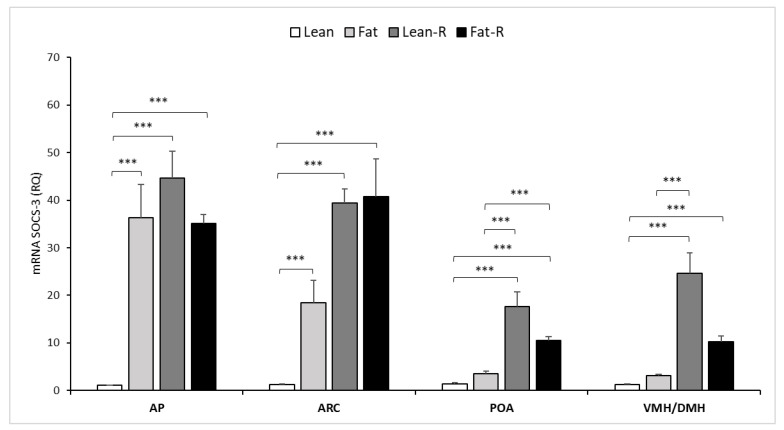
Expression of SOCS-3. The mean expression (±SEM) of a suppressor of cytokine signaling-3 (SOCS-3) mRNA in the ovine anterior pituitary (AP), arcuate nucleus (ARC), preoptic area (POA) and ventro- and dorsomedial nuclei (VMH/DMH) collected during long-day (LD) photoperiod in nontreated (Lean and Fat) animals and animals treated with recombinant bovine resistin (Lean-R [Lean-Resistin treated] and Fat-R [Fat-Resistin treated]. The expression of SOCS-3 mRNA is reported in arbitrary units (RQ) relative to cyclophilin mRNA and expressed relative to the calibrator sample. Differences relative to the control or between the other groups are indicated with ** *p* < 0.05 or *** *p* < 0.001.

**Table 1 ijms-21-04238-t001:** Plasma concentrations (mean ± SEM) of metabolic parameters (resistin, leptin, insulin, glucose, nonesterified fatty acids (NEFAs), cholesterol, triglycerides, high-density lipoprotein (HDL) and low-density lipoprotein (LDL) in nontreated (Lean and Fat) animals and animals treated with recombinant bovine resistin (Lean-R [Lean-Resistin treated] and Fat-R [Fat-Resistin treated]; *n* = 5) after 5 months of body weight alterations (*n* = 5 per group).

Plasma Hormones and Metabolites	Lean	Fat	Lean-R	Fat-R	a vs. b	c vs. d	a vs. c	b vs. d
Average resistin plasma concentrations (ng/mL)	5.4 ± 0.2	9.32 ± 0.7	6.44 ± 0.8	14.87 ± 1.2	*	*	*	*
Average leptin plasma concentrations (ng/mL)	2.61 ± 0.13	6.49 ± 0.35	4.69 ± 0.14	17.53 ± 0.53	**	***	**	***
Average insulin plasma concentrations (µU/mL)	2.14 ± 0.32	10.55 ± 2.27	4.97 ± 1.69	91.07 ± 27.27	**	***	ns	***
Average glucose plasma concentration (mmol/L	3.12 ± 0.07	4.60 ± 0.09	2.97 ± 0.05	5.83 ± 0.08	ns	*	*	*
Average NEFA plasma concentrations (µmol/L)	394.51 ± 23.7	635.33 ± 32.5	422.61 ± 45.4	560.34 ± 35.1	**	**	ns	**
Cholesterol (mg/dL)	36.70 ± 0.45	45.26 ± 1.72	32.74 ± 1.83	38.78 ± 2.44	ns	*	*	*
Triglycerides (mg/dL)	10.06 ± 0.33	13.80 ± 2.48	16.29 ± 1.84	19.83 ± 3.77	**	*	*	*
HDL fraction (mg/dL)	13.03 ± 0.44	18.87 ± 4.39	10.87 ± 0.19	13.90 ± 0.18	*	ns	*	ns
LDL fraction (mg/dL)	21.25 ± 0.36	19.60 ± 1.16	19.19 ± 2.14	20.92 ± 1.53	ns	ns	ns	ns

Abbreviation: HDL, high-density lipoprotein; LDL, low-density lipoprotein; ns, not significant. a, denotes, lean; b, denotes, fat; c, denotes, Lean-R; d, denotes, Fat-R. Values are means (±SEM; *, *p* < 0.05; **, *p* < 0.01; ***, *p* < 0.001).

**Table 2 ijms-21-04238-t002:** Plasma concentrations (mean ±SEM) of hormones (LH, FSH, PRL and GH) ) in nontreated (Lean and Fat) animals and animals treated with recombinant bovine resistin (Lean-R [Lean-Resistin treated] and Fat-R [Fat-Resistin treated]; *n* = 5) after alter 5-month with body weight alternation (*n* = 5 per group).

Plasma Hormones	Lean	Fat	Lean-R	Fat-R	a vs. b	c vs. d	a vs. c	b vs. d
Average plasma LH concentration (ng/mL)	2.6 ± 0.3	4.2 ± 0.8	3.6 ± 0.2	4.8 ± 0.1	**	*	**	**
LH pulse amplitude (ng/mL)	2.7. ± 0.5	3.09 ± 0.8	3.3 ± 0.2	3.8 ± 0.2	*	*	**	**
Average plasma FSH concentrations (ng/mL)	4.5 ± 0.8	6.2 ± 0.6	7.9 ± 0.9	6.9 ± 0.5	*	ns	*	*
Average plasma PRL concentrations (ng/mL)	65.1 ± 3.5	76.6 ± 4.3	123 ± 5.6	174 ± 7.9	*	**	*	**
Average plasma GH concentration (ng/mL)	14.35 ± 0.21	8.13 ± 0.25	12.14 ± 0.734	2.68 ± 0.14	***	**	**	**

Abbreviation: LH, luteinizing hormone; FSH, follicle stimulating hormone; PRL, prolactin; GH, growth hormone; ns, not significant. a, denotes, lean; b, denotes, fat; c, denotes, Lean-R; d, denotes, Fat-R. Values are means (± SEM; *, *p* < 0.05; **, *p* < 0.01; ***, *p* < 0.001).

**Table 3 ijms-21-04238-t003:** Characteristics of primers and probes used to determine cyclophilin (CPH; reference gene), the long form of the leptin receptor (LRb; target gene) and a suppressor of cytokine signaling-3 (SOCS-3; target gene) mRNA expression in sheep.

Gene	Primer Sequence (5′–3′)	Probe Sequence (5′–3′)	Amplicon Size	GenBank Accession Number
CPH	CGGCTCCCAGTTCTTCATCA	FAM-CGTTCCGACTCCGC-MGB	64 bp	D14074
	ACTACGTGCTTCCCATCCAAA			
LRb	CGACGAGGGTGGCATATTTAA	FAM-CAGGAGACAGCCCTC-MGB	63 bp	U62124.1
	CAGACATAACCTGTGAGGATGGAA			
SOCS-3	CCTCAAGACCTTCAGCTCCAA	FAM-AGCGAGTACCAGCTGG-MGB	68 bp	NM_174466

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
