# Peer review of "Changes in Expression of the Genes for the Leptin Signaling in Hypothalamic-Pituitary Selected Areas and Endocrine Responses to Long-Term Manipulation in Body Weight and Resistin in Ewes"

_ijms, 2020, doi:10.3390/ijms21124238_

Round 1
Reviewer 1 Report
The manuscript of Zieba et al. summited to the International Journal of Molecular Sciences describes the role of resistin (RSTN) in leptin signaling and endocrine profile in ewes with different body adiposity. The authors concluded “that nutritionally altered expression of LRb and SOCS-3 and concentrations in hormonal parameters can be linked with leptin resistance and endocrine and metabolic adaptations that occur with long-term changes in adiposity.”
General comments
This manuscript brings new information concerning the role of RSTN in regulation of selected hormones’ secretion and expression of selected genes in ewes with different body adiposity. This is a novel study, and this area of research warrants further investigation. The methodology is adequate to accomplish research goals, and the manuscript is well prepared for publication. However, the data presentation and description need some improvements.
Specific comments
- L1. I suggest improving the title.
- L21. Please use the term “administration” or “injection”.
- L28. It should read “We suggest that nutritionally…”
- L29. The term “concentrations in hormonal parameters” is confusing, please correct.
- L58-72. Please rewrite these sentences for clarification. Several sentences are too long and confusing; please correct.
- L91-99. Please provide the hypothesis for this study, and clearly list all objectives.
- L100. Throughout the Results section please delete “(mean ± SEM)” since this information is provided for each table.
- L102. Please specify time when BW was recorded.
- L104. “Less” but not “lower”, and please specify time.
- L107. Please mark statistical differences in this figure.
- L110-11. Please specify time.
- L112-114. Please combine these sentences.
- L119 and 134. The marking of statistical differences is extremely confusing in Tables 1 and 2. I suggest using small letters “a, b, c, d” to show that four values are statistically different (or similar) within a row. You can provide the P value as “P<0.001-0.05” that clearly demonstrates valid statistical differences.
- L157. Please add “mRNA” after “expression”.
- L163. Please add “mRNA” before “expression”.
- L221. Please replace “domesticated” with “domestic”.
- L338. Please provide time of the year when this experiment was conducted.
- L450 and 474. Please replace “Molecular” with “Quantitative real time RT-PCR”.
Overall, this is an interesting study, worth publishing, but requires revision and some improvements.
Author Response
Reviewer # 1
We took into account all comments when revising the manuscript.
A detailed list of small changes is presented below, and we indicated changes directly in the text using red font.
Specific points:
- I suggest improving the title.
We agree with the Reviewer, the title should be improved: “Changes in Expression of the Genes for the Leptin Signaling in Hypothalamic-Pituitary Selected Areas and Endocrine Responses to Long-Term Manipulation in Body Weight and Resistin in Ewes”.
- L 21. Please use the term “administration” or “injection”.
The term “treated” has been changed for “administration” according to Reviewer’s suggestion.
- 28. It should read “We suggest that nutritionally…”
The change has been made improving the sentence according to Reviewer’s suggestion.
- 29. The term “concentrations in hormonal parameters” is confusing, please correct.
We agree that it is confusing term, we suggest to change for “reproductive and metabolic hormones concentrations”.
- L. 58-72. Please rewrite these sentences for clarification. Several sentences are too long and confusing; please correct.
Reviewer is right, the sentences were confusing and too long. They have been shortened, making them clearer for readers.
Earlier, in 2018, our team presented that in seasonally breeding ovariectomized ewes with estradiol replacement action of RSTN on the level of AP appeared to be firmly dose- and photoperiod-dependent. RSTN modulated reproductive hormone secretions: luteinizing hormone (LH), follicle stimulating hormone (FSH) and prolactin (PRL) [11]. Thus, it has been suggested that RSTN is engaged in the regulation of seasonal reproduction in sheep [11].
Nutrition can alter the expression of many peptides and neuropeptides that regulate food intake. However, most experiments in animals have investigated acute changes with fasting [12,13], and the effects of long-term alterations in body weight and adiposity still require complete explanation.
- 91-99. Please provide the hypothesis for this study, and clearly list all objectives.
We agree with Reviewer that the description of our objectives was quite messy. We changed sentences to provide clear hypothesis and objectives.
In the current study, we used two different metabolic/nutritional sheep models (Lean and Fat) to test the hypothesis that alternations in body weight affect RSTN-mediated effects on leptin signaling component expressions (LRb and SOCS-3) at the level of the selected hypothalamic nuclei (ARC, medio-basal hypothalamus comprising ventro-medial/dorso-medial – VMH/DMH nuclei, preoptic area [POA]) and AP, brain areas engaged directly in the regulation of metabolism and leptin action. Furthermore, the endocrine (reproductive and metabolic) indicators in Lean and Fat sheep with alternate leptin concentrations (low and high), were investigated.
- 100. Throughout the Results section please delete “(mean ± SEM)” since this information is provided for each table.
“(mean ± SEM)” has been deleted throughout the whole Results section.
- 102. Please specify time when BW was recorded.
In M&M section L. 358-359 is stated that sheep were weighted every second week. However, we added information about the specific time when the differences in body weight occurred.
The body weights of the two groups of sheep was not significant within every second week of weighting. At the time of sampling, there was significant (P<0.01) difference in body weights of the two groups..
- “Less” but not “lower”, and please specify time.
Change has been made.
- Please mark statistical differences in this figure.
We explained that issue in point 7.
- 110-111. Please specify time.
The estradiol implants were prepared by myself according to dr. Marcel Amstalden instruction (Texas A&M University) based on references by Karsch and Foster 1975. Endocrinology 97:373; Day et al. 1984.Biol. Reprod. 31:332 and Gazal et al. Biol. Reprod. 59:676.
Materials for implants were: Silastic® Brand Tubing: 0.132 in. (3.35 mm) ID x 0.183 in. (5.65 mm) OD (Dow Corning, cat.# 601-335), 17β Estradiol (Sigma, Cat#E8875), Silicone (Medical Adhesive silicone) (Dow Corning Cat. # 891; Applied Silicone Corporation Cat. #40064).
We used those implants frequently and they provide 17β-estradiol concentration between 3.6-4.0 pg/ml, as we determine by enzyme immunoassay (EIA) kits (DRG Instruments GmbH, Marburg, Germany) according to the manufacturer’s instructions, e.g.:
Ewes were ovarectomised about 45 days before the study began, to eliminated variation due to the differences in the concentrations of sex hormones and about 10 days after ovarectomy estradiol implants were inserted.
To check the proper estradiol implants function, we determined estradiol concentration just before experiments with RSTN started in all groups of sheep.
- L 112-114. Please combine these sentences.
The sentences have been combined.
The concentrations of circulating RSTN, leptin, insulin, and NEFA were lower (P < 0.05) for RSTN and (P<0.01) for the other hormones in the Lean vs Fat groups of ewes (Table 1).
- 119 and 134. The marking of statistical differences is extremely confusing in Tables 1 and 2. I suggest using small letters “a, b, c, d” to show that four values are statistically different (or similar) within a row. You can provide the P value as “P<0.001-0.05” that clearly demonstrates valid statistical differences.
In order to satisfied the two reviewers, we decided to leave tables as they are presented in manuscript for two reasons: 1. Reviewer # 2 wrote that “these results have been presented with appropriate tables and figures..” and 2. we want to present the results in the same way in tables and figures by showing significance with proper number of asterisks for P value.
- L.157. Please add “mRNA” after “expression”.
The “mRNA” has been added according to Reviewer suggestion.
- L163. Please add “mRNA” before “expression”.
The “mRNA” has been added according to Reviewer suggestion.
- L221. Please replace “domesticated” with “domestic”.
The replacement has been made according to Reviewer suggestion.
- L338. Please provide time of the year when this experiment was conducted.
We specified the time of experiments.
“during LD season (between April-May).
- L450 and L474 Please replace “Molecular” with “Quantitative real time RT-PCR”.
The replacements have been made according to Reviewer suggestion.
Reviewer 2 Report
The manuscript "Resistin and Alternations in Body Weight Change the Expression of Leptin Signaling Genes and Endocrine Responses in Ewes" of Zieba et al. aimed to know how long-term metabolic manipulation affects
resistin-mediated effects on leptin signaling component expression at the level of specific selected hypothalamic nuclei (ARC, medio-basal hypothalamus comprising ventro medial/dorso-medial – VMH/DMH nuclei, preoptic area [POA]) and AP, brain areas engaged directly in the regulation of metabolism and leptin action.
The contribution shows that the alteration in body weight influences the effects of resistin on metabolism and reproductive parameters in sheep and the leptin signaling pathway. In particular, it proposes that nutritionally altered expression of leptin receptor b and of suppressor of cytokine signaling-3 can be linked with the endocrine and metabolic adaptations. Resistin regulates the release of reproductive hormones from the pituitary and metabolic processes of sheep.
These results have been presented with appropriate tables and figures and it was discussed in details to justify the merit of the study. The previous studies of the authors are relevant and the bibliography of this manuscript is valid. The paper shows that the author's knowledge in English is good.
Finally, I suggest few changes and important integrations reported above.
The last sentence in the abstract needs remodulation to be more clear.
Line 92-94 in the introduction can be deleted.
It is reported that to determine the expression levels of a suppressor of cytokine signaling-3 and the long form of the leptin receptor you have determined in selected brain regions, such as the anterior pituitary, hypothalamic arcuate nucleus, preoptic area, and ventro- and dorsomedial nuclei. Please report briefly how you detect anatomically these areas collected.
In line 102 you report "bodyweights" in place od "body weights" .Table 1 report the same problem.
Line 108-109 can be deleted. Remain please short indication about P value.
Try to delete old references already present in your previous manuscripts adding (for reference see Zieba et al. ****).
I suggest to remain from 2000 to now and to add more relevant in the last few years. Check properly in the journal selected for this manuscript.
Author Response
Reviewer # 2
We took into account all comments when revising the manuscript.
A detailed list of small changes is presented below, and we indicated changes directly in the text using red font.
Specific points:
- The last sentence in the abstract needs remodulation to be more clear.
….metabolic and reproductive hormones concentrations and the expression of leptin signaling components: LRb and SOCS-3. This may be an adaptive mechanism to long-term changes in adiposity during the state of leptin resistance.
- Line 92-94 in the introduction can be deleted.
In order to satisfied the two reviewers, we combined two suggestions and tried to change these sentences to get the best solution.
In the current study, we used two different metabolic/nutritional sheep models (Lean and Fat) to test the hypothesis that alternations in body weight affect RSTN-mediated effects on leptin signaling component expressions (LRb and SOCS-3) at the level of the selected hypothalamic nuclei (ARC, medio-basal hypothalamus comprising ventro-medial/dorso-medial – VMH/DMH nuclei, preoptic area [POA]) and AP, brain areas engaged directly in the regulation of metabolism and leptin action. Furthermore, the endocrine (reproductive and metabolic) indicators in Lean and Fat sheep with alternate leptin concentrations (low and high), were investigated.
- It is reported that to determine the expression levels of a suppressor of cytokine signaling-3 and the long form of the leptin receptor you have determined in selected brain regions, such as the anterior pituitary, hypothalamic arcuate nucleus, preoptic area, and ventro- and dorsomedial nuclei. Please report briefly how you detect anatomically these areas collected.
The following sentences have been added:
The selected brain regions were collected by removing a tissue block encompassing the hypothalamic–infundibular complex, followed by transection into two halves. An anterior coronal cut was made ~3–5 mm rostral to the optic chiasm, and a posterior coronal cut was made, which contained approximately one-third of the mamillary body. A longitudinal cut parallel to the ventral surface of the brain ~2–3 cm dorsal to the anterior commissure followed. At the same time, the pituitary was harvested from the sella turcica.
- In line 102 you report "bodyweights" in place of "body weights" .Table 1 report the same problem.
We corrected the problem.
- Line 108-109 can be deleted. Remain please short indication about P value.
In order to satisfied the two reviewers, we combined two suggestions and tried to change these sentences to get the best solution.
- Try to delete old references already present in your previous manuscripts adding (for reference see Zieba et al. ****).
We deleted refs. 36 and 37 as per Reviewer suggestion. It is quite difficult to find more references concerning leptin and resistin interactions. We are the second group (one is Reverchon et al.) working on ruminant model and conducting in vivo experiments using RSTN.
As for leptin study we left References for papers published before 2000 by Zhang et al., Mercer et al., Kasim-Karakas et al., Dyer et al, Thissen et al. Those scientists are pioneers in leptin experiments and studies on seasonal animals.